# Energy-Shaped Manifold Projections Enable Adversarial Detection

**Artem Matevosian**
Laboratory of Innovative Technologies for Processing Video Content, Innopolis University
Innopolis, Russia
a.matevosian@innopolis.university

**Bader Rasheed**
Research Center of the Artificial Intelligence Institute, Innopolis University
Innopolis, Russia
b.rasheed@innopolis.university

## Abstract

Adversarial attacks and distribution shift undermine reliability of deep classifiers. We revisit energy-based out-of-distribution (OOD) detection and propose a simple projection head that maps representations onto a learned data manifold and uses the squared norm of the projected vector as an energy score. The training is parallel with classification loss on the classification head and soft energy separation loss on the projection head that pushes adversarial examples to high energy while keeping clean examples at low energy. On a CIFAR-10 (Krizhevsky [2009]) variant with a held-out 10th class acting as OOD, our method detects fast gradient sign (FGSM), projected gradient descent (PGD), and AutoAttack (AA) adversarial examples even when the classifier remains non-robust. We study design choices, including hinge versus softplus energy losses, regularization on the projected vector, and the importance of normalization layer choice to align train and test statistics. Despite energy separation transferring across attacks, we find little OOD rejection of unrelated images and highlight failure modes. Our work provides a critical analysis of energy-shaped projections and outlines open problems and possibilities for future research.

## 1 Introduction

Machine learning systems deployed in high-stakes applications must cope with unreliable data: inputs may be perturbed by adversaries, drawn from shifted distributions, contain missing or biased values, or arise from human interaction. Standard training objectives optimize for accuracy, but offer no guarantees when inputs deviate from the training distribution. Recent work emphasizes OOD input detection as a complementary strategy to robust classification.

In this paper, we revisit energy-based detection for adversarial perturbations and present the energy-shaped manifold projection head. The method maps the last hidden representation $z$ from a standard backbone to a lower- or the same-dimensional representation $z'$; the squared norm $E = \|z'\|^2$ is used as an energy score. The soft separation loss encourages low energy for clean examples and high energy for adversarial data while classification is trained in parallel. We implement flexible loss functions (ReLU (hinge), softplus, and squared hinge) and add $L_2$ regularization on $z'$ to prevent the magnitude explosion. Training uses FGSM for efficiency and separates gradients flowing through the energy and classification heads to the adversary, preventing energy awareness.

39th Conference on Neural Information Processing Systems (NeurIPS 2025) Workshop: Reliable ML from Unreliable Data.

Despite its simplicity, our energy head detects adversarial examples produced by stronger (PGD, AA) attacks and does not react to natural OOD data, provided that batch-independent normalization is used, so that training and evaluation compute energies consistently. However, we also observe the limitations: the classification robustness does not transfer and the energy values can explode when the hinge loss is used without regularization.

**Contributions** (i) We propose a projection head yielding an energy score $E = \|z'\|^2$. (ii) We introduce the soft energy separation loss with $L_2$ regularization and analyze its stability. (iii) We implement FGSM training and FGSM+PGD-20+AA evaluation on CIFAR-9 with the 10th class as OOD, reporting AUROC and robust-after-rejection metrics. (iv) We demonstrate that batch-independent normalization is crucial for energy alignment between training and testing. (v) We demonstrate that our method does not mistake OOD for adversarial data. (vi) We report failure cases, such as non-transfer of classification robustness, and provide the details on head complexity and loss functions.

## 2 Related Work

**Energy-based OOD detection.** Liu et al. [2020] propose using the energy defined by the negative log partition function as a score for OOD detection and show that it reduces the false positive rate by 18% compared to the softmax confidence. Their framework allows energy to be used as a parameter-free inference score or as a trainable cost function with square hinge loss. We adapt such an idea, but use the squared norm of a projection instead of the logit-based energy and train the projection head jointly with classification.

**Adversarial training and attacks.** Adversarial training casts robustness as a saddle-point optimization problem and uses the inner maximization to generate worst-case perturbations. Madry et al. [2018] identify projected gradient descent (PGD) as a universal first-order adversary and demonstrate robust models on MNIST and CIFAR. The fast gradient sign method (FGSM) introduced by Goodfellow et al. [2015] provides an efficient way to generate adversarial examples by linearizing the loss around the input. Our training uses energy-blind FGSM, while evaluation includes FGSM, PGD-20, and AutoAttack. AutoAttack (AA, Croce and Hein [2020]) combines multiple attacks to reliably evaluate robustness and highlights that PGD may overestimate robustness; it recommends using an ensemble of attacks instead.

**Normalization layers and dataset shift.** Batch normalization (BN, Ioffe and Szegedy [2015]) normalizes layer inputs using batch statistics to reduce internal covariate shift, improving training speed and acting as a regularizer. However, BN uses running estimates during evaluation, and mismatched statistics under distribution shift can harm performance. We find that using batch-independent normalization (e.g. instance normalization, Ulyanov et al. [2016]) is necessary to align energy distributions between train and test.

**Uncertainty and distribution shift.** Robustness under distribution shift and OOD inputs is necessary for safe deployment. Ovadia et al. [2019] benchmark predictive uncertainty methods and show that calibration in the i.i.d. setting does not translate to calibration under shift and that evaluating uncertainty under shift is more meaningful. Our method complements this line by focusing on detection via energy scores rather than calibration.

## 3 Method

### 3.1 Architecture and Energy Score

Let $f_\theta$ denote the backbone mapping an input image $x \in \mathbb{R}^d$ to a representation $z = f_\theta(x)$. We append a projection head $f_\eta$ that maps $z$ to the same or lower dimensional vector $z' = f_\eta(z) \in \mathbb{R}^k$. The classifier branch predicts the class probabilities from $z$ via a linear layer and cross-entropy loss. The energy branch computes the score $E(z') = \|z'\|_2^2$ that we aim to make small for clean inputs and large for adversarial ones. In practice, $f_\eta$ is a small multilayer perceptron.

## 3.2 Energy Separation Loss

Given a batch of clean examples $\{x_i\}$ and adversarial examples $\{x_i^{\mathrm{adv}}\}$ generated on the fly, we compute energies $E_i$ and $E_i^{\mathrm{adv}}$ as described above. We minimize the total loss

$$\mathcal{L} = \frac{1}{B} \sum_i \Big[ \underbrace{\mathrm{CE}(y_i, f_\theta(x_i))}_{\text{classification}} + \lambda_{\mathrm{sep}}\, \ell_{\mathrm{sep}}\big(E_i, E_i^{\mathrm{adv}}\big) + \lambda_2 \|z_i'\|_2^2 \Big], \tag{1}$$

where CE is the cross-entropy loss and $\ell_{\mathrm{sep}}$ encourages separation between clean and adversarial energies. We experiment with three variants:

- **Hinge loss:** $\ell_{\mathrm{sep}}(E, E^{\mathrm{adv}}) = \max(0, \epsilon - E) + \max(0, E^{\mathrm{adv}} - (\epsilon + \Delta))$. This penalty leads to energy explosion in practice unless $\lambda_2$ is tuned.
- **Softplus:** $\ell_{\mathrm{sep}}(E, E^{\mathrm{adv}}) = \mathrm{softplus}(\epsilon - E) + \mathrm{softplus}(E^{\mathrm{adv}} - (\epsilon + \Delta))$, which is differentiable and alleviates gradient vanishing during training, making the joint training process more stable.
- **Squared hinge:** $\ell_{\mathrm{sep}}(E, E^{\mathrm{adv}}) = c \cdot \max(0, \epsilon - E)^2 + c \cdot \max(0, E^{\mathrm{adv}} - (\epsilon + \Delta))^2$ that behaves similarly to softplus, provided that $c$ is small enough to prevent initial penalty explosion.

We set $\epsilon$ as the maximum allowed value for clean energy and $\Delta$ as a margin hyperparameter. Regularization on $z'$ prevents the projection from shrinking or exploding. During the adversarial example generation, we do *not* backpropagate through the energy branch, ensuring the attack is *energy-blind* and does not exploit our detector.

## 3.3 Adversarial Training and Evaluation

Adversarial training solves a saddle-point problem in which the inner maximization generates adversarial perturbations and the outer minimization updates the model parameters. We use FGSM for its efficiency and backpropagate only through the classification branch. The perturbations are constrained in the $\ell_\infty$ norm ball with radius $\varepsilon = 8/256$.

During evaluation, we generate adversarial examples using FGSM ($\varepsilon = 16/256$), 20-step PGD ($\varepsilon = 16/256$) with step size $\alpha = 2/256$, and AutoAttack ($\varepsilon = 8/256$). Then, we calculate area under the ROC curve (AUROC). We also report robust-after-rejection accuracy: classification accuracy over all the examples that did not exceed $\epsilon + \frac{\Delta}{2}$. OOD experiments treat the 10th class in CIFAR-10 as unknown and evaluate whether the energy rejects these inputs.

## 3.4 Normalization Alignment

During preliminary experiments, we observed that energy distributions for clean and adversarial examples behave differently between training and evaluation, often collapsing or even reversing. Investigation revealed that our backbone used batch normalization layers that adapt to batch statistics during training but use running estimates at evaluation. When adversarial examples dominated the batch, the running statistics drifted and corrupted the energy. To remedy this, we use instance normalization to perform exactly the same calculations both in train and test time. Figure 1 illustrates how using IN stabilizes energy distributions.

## 4 Experiments

We implement our method in PyTorch (https://github.com/ArtMGreen/manifold-projection-layer). The backbone is a pre-classification head ResNet-18 trained on CIFAR-9, i.e. CIFAR-10 without class 10 ("truck"); the removed class serves as OOD data. The images are normalized and no data augmentation is used. We train for 5 epochs with batch size 16 using stochastic gradient descent with momentum 0.9 and learning rate 0.02. FGSM attacks use $\varepsilon = 8/255$. The hyperparameters $\lambda_{\mathrm{sep}}$ and $\lambda_2$ are tuned to facilitate smooth joint training without energy explosion or any part of loss dominating the training regime; we typically set $\lambda_{\mathrm{sep}} = 1$ and $\lambda_2 = 5 \cdot 10^{-3}$. For evaluation, we generate 9,000 adversarial examples for each attack type and compute accuracy on clean and adversarial examples, AUROC in adversarial and OOD detection, and robust-after-rejection accuracy at the rejection threshold $\epsilon + \frac{\Delta}{2}$.

# 5 Results and Analysis

**Detection versus classification** Table 1 summarizes the results. Energy-shaped projections achieve near-perfect AUROC for both FGSM and PGD attacks ($> 0.99$) and high AUROC under AutoAttack ($> 0.84$), even when the classification accuracy on the examples is zeroed. This indicates that adversarial perturbations cause a predictable increase in the energy norm even if the classifier fails on the perturbed images. Therefore, energy separation transfers to unseen attacks, but the classification robustness does not: a stronger adversary manages to nullify classification accuracy.

Table 1: Detection performance on CIFAR-9 test set. $AUROC_{adv}$ and $AUROC_{OOD}$ are measured for adversarial vs clean and OOD vs clean detection tasks respectively; $ACC_{clean}$ and $ACC_{adv}$ denote classification accuracy on clean and adversarial examples respectively; RAR is robust-after-rejection accuracy at $\epsilon + \frac{\Delta}{2}$ rejection threshold.

| Adversary | $ACC_{clean}$ | $ACC_{adv}$ | $AUROC_{adv}$ | $AUROC_{OOD}$ | RAR |
|---|---|---|---|---|---|
| FGSM | 0.6886 | 0.7821 | 1 | 0.5522 | 0.6886 |
| PGD-20 | 0.6886 | 0 | 0.9976 | 0.5522 | 0.6597 |
| AutoAttack | 0.6886 | 0 | 0.8427 | 0.5522 | 0.525 |

**OOD detection** When evaluating on the held-out CIFAR class, energy scores for OOD images closely match those of clean in-distribution examples. The AUROC for OOD versus clean detection is around 0.55, indicating near-random performance. Therefore, while energy-shaped projections do not replace standard OOD detection mechanisms, they might be compatible with these, since OOD data is not mistaken for adversarial. Figure 2 provides ROC curves as an illustration.

# 6 Limitations and Broader Impact

Our study has several limitations. First, we evaluate on CIFAR-like data; the results may not generalize to more complex domains or modalities. Second, training uses FGSM; while detection transfers to PGD and AutoAttack, we have not evaluated energy-aware attacks, which might circumvent our detector. Third, the projection head is tuned manually; automating its architecture and hyperparameters is left to future work. Finally, our method does not address distribution shift beyond adversarial perturbations: energy is not designed detect unrelated OOD inputs. We encourage future work to evaluate compatibility of other OOD detection methods with energy-based projection heads.

# 7 Conclusion

We proposed an energy-shaped manifold projection head for adversarial detection. By training a projection head with a soft separation loss and regularizing the projected representation, we obtain a robust energy score that distinguishes adversarial inputs even when the classification robustness fails. Our experiments highlight the importance of normalization layer choice and show that softplus and squared hinge losses provide stable energy separation. At the same time, we report negative results: the method does not reject OOD data unrelated to the training distribution, and classification robustness does not improve. We hope our analysis and ablations will inspire further research into reliable detection mechanisms.

# 8 Acknowledgements

The study has been supported by the Ministry of Economic Development of the Russian Federation (agreement No. 139-10-2025-034 dd. 19.06.2025, IGK 000000C313925P4D0002)

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

# A  Figures and Illustrations

## A.1  Energy drift under batch-dependent normalization

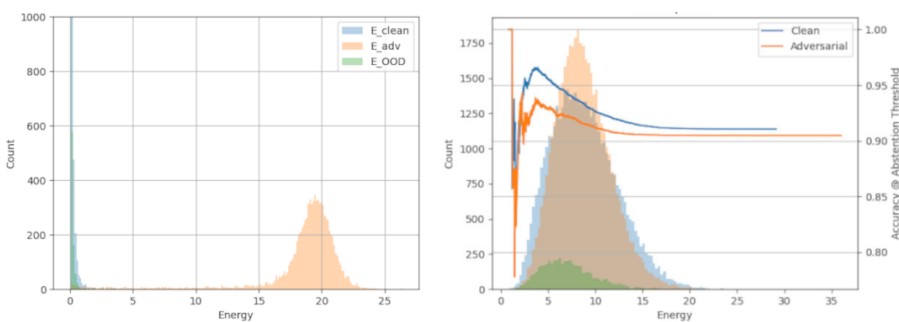

Figure 1: Energy histograms with instance (left) and batch (right) normalization. Under BN, clean and adversarial energies overlap. Using IN shifts adversarial energies higher and clean energies lower, enabling separation.

## A.2  Threshold sweep and rejection

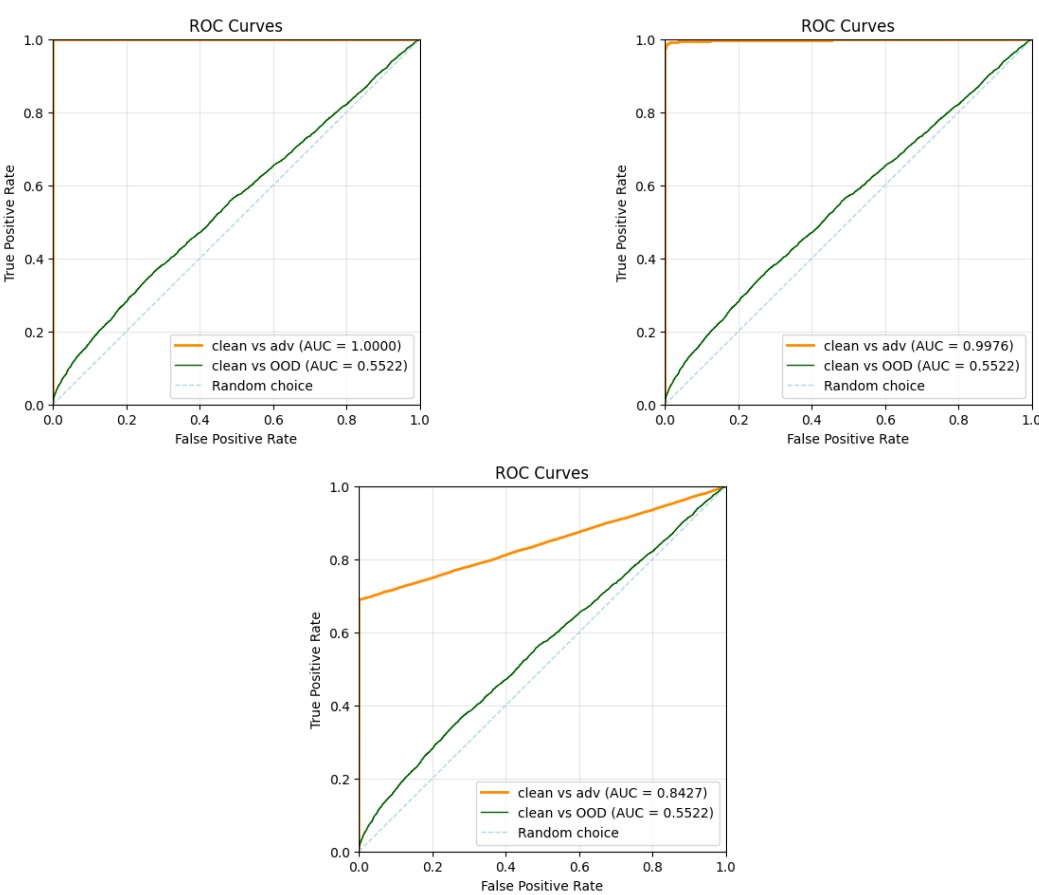

Figure 2: ROC curves for FGSM (left), PGD (right), and AA (bottom) attacks. Performance under FGSM and PGD-20 is close to perfect. Performance under AutoAttack is not perfect anymore but remains high. Energy-trained model is consistently good in adversarial detection, but is not suitable for OOD detection.

