# OpenReview forum: "Energy-Shaped Manifold Projections Enable Adversarial Detection"
_NeurIPS.cc/2025/Workshop/Reliable_ML — NeurIPS 2025 - Reliable ML Workshop_

### Official Review · Reviewer_KKWT · 2025-09-16
**Incremental Novelty with Insufficient Experimental Breadth**

**Rating:** 6
**Confidence:** 3

**Review:**

### **Summary**
The paper introduces Energy-Shaped Manifold Projections as a method for adversarial detection. A projection head maps backbone representations to a lower-dimensional space, and the squared norm of the projected vector is used as an energy score. Training jointly optimizes classification and a soft energy separation loss that enforces low energy for clean data and high energy for adversarial examples. Experiments are conducted on CIFAR-9 (CIFAR-10 with one class held out as OOD). Results show near-perfect AUROC for FGSM and PGD adversarial detection. However, the method does not reject unrelated OOD samples effectively and fails to improve classification robustness. The authors also analyze normalization choices and loss functions, providing a careful ablation study results.

### **Strength**
* Novelty: The main novelty lies in the projection-based formulation and stability analysis. This makes the contribution incremental but still useful and clearly presented.
* Clarity and Rigor: The method is described clearly, with detailed loss definitions and ablations.
* Empirical Quality: Strong adversarial detection performance.
* Relevance: Tackles adversarial detection under unreliable data — closely relevant topic.

### **Weakness/Limitations**
* OOD Generalization: Still fails on natural OOD inputs, suggesting limited practical deployment.
* Evaluation Scope: Only CIFAR-9/10 experiments; unclear scalability.
* Adversary Coverage: No evaluation with stronger/adaptive attacks (e.g., AutoAttack).
* Statistical Limitation: No error bars or compute details provided.
* Robustness Transfer: Classification robustness does not improve, restricting broader applicability.

## **Suggestions for Authors**
* Strengthen evaluation: Include AutoAttack/adaptive adversaries.
* Hybrid approaches: Explore combining projection-based energy with established OOD detection to cover both adversarial and natural shifts.
* Reproducibility: Add statistical error analysis and compute resource details.
* Provide broader experimental coverage: for example, testing on larger or more diverse datasets (ImageNet subsets, non-vision domains) to strengthen generality.

---

### Official Review · Reviewer_e1MN · 2025-09-17
**Good paper with clear contributions**

**Rating:** 7
**Confidence:** 3

**Review:**

Summary
The paper introduces an energy-shaped manifold projection head designed for adversarial detection in deep classifiers. The central idea is that even when the classifier fails to correctly predict the label of an adversarial example, the model can still output robust energy scores that help distinguish adversarial inputs from benign ones. To achieve this, the authors train a projection head with a combination of a soft separation loss and a regularization term that enforces separation in the projected space. Through a series of experiments, they demonstrate that certain loss functions—particularly softplus and squared hinge losses—are effective in providing stable energy separation, yielding consistent performance across multiple adversarial attack settings. At the same time, the experiments reveal a limitation of the approach: the method struggles to reject out-of-distribution (OOD) data that is unrelated to the training distribution. This shortcoming suggests that their method should not be viewed as a complete substitute for OOD detection but rather as a complementary tool that focuses specifically on adversarial robustness.

Strengths
A major strength of this work is that it thoughtfully integrates ideas from several different but related research threads—namely, energy-based OOD detection methods, adversarial training, and adversarial attack strategies—into a single coherent framework for adversarial detection. The novelty lies not only in the introduction of the energy-shaped projection head but also in the careful design of the training objectives, which allow the projection space to be structured in a way that highlights adversarial anomalies. Another strength is the clarity of presentation: the experiments are explained logically, the results are reported transparently, and the authors openly discuss both the successes and the limitations of their approach. This honesty is commendable and gives the paper credibility. Furthermore, the work directly addresses the critical problem of adversarial data detection—a problem of growing importance as machine learning systems are increasingly deployed in real-world contexts where data is imperfect and adversaries may be present. The methodological contribution is therefore not only of theoretical interest but also of practical relevance.

Weaknesses and Limitations
The authors themselves are forthright in acknowledging the limitations of their work, and I will echo those points here. First, while the method is effective for distinguishing adversarial examples, it does not provide a robust solution for OOD detection. This limitation means that the approach must be complemented with separate OOD detection techniques if the system is to be reliable in real-world environments. Second, the experiments are conducted primarily on CIFAR-like datasets, raising questions about whether the findings will generalize to more complex domains. Third, because the method relies on energy separation, it may itself be vulnerable to more sophisticated attacks which are designed to exploit weaknesses in energy-based defenses. Finally, the projection head introduces additional design complexity: its architecture and hyperparameters require careful tuning, and automating this process remains an open challenge. These limitations do not undermine the contribution of the paper but do suggest important avenues for future work.

Suggestions for the Authors
While the authors have already offered a fair and balanced discussion of limitations, one direction that could be valuable is to explicitly explore hybrid approaches that combine their projection head with standard OOD detection techniques. Such an exploration would illustrate more concretely the complementary relationship between the two approaches. Additionally, extending the experiments to a wider set of domains and testing against adaptive, energy-aware adversaries would help strengthen the claims of robustness. Finally, investigating automated or principled methods for tuning the projection head could further enhance the practicality of the proposed framework.

Ethics
The paper does not raise any ethical concerns. The contributions are methodological and empirical in nature, and the authors are transparent about both the strengths and weaknesses of their approach. The work is aligned with improving the reliability of machine learning systems, which is generally a positive step in terms of ethical and societal impact.